# Chromium Methionine and Ractopamine Supplementation in Summer Diets for Grower–Finisher Pigs Reared under Heat Stress

**DOI:** 10.3390/ani13162671

**Published:** 2023-08-19

**Authors:** Esther Sánchez-Villalba, Eileen Aglahe Corral-March, Martín Valenzuela-Melendres, Libertad Zamorano-García, Hernán Celaya-Michel, Andrés Ochoa-Meza, Humberto González-Ríos, Miguel Ángel Barrera-Silva

**Affiliations:** 1Departamento de Agricultura y Ganadería, Universidad de Sonora, Carretera a Bahía de Kino Km. 21, Hermosillo 83000, Mexico; esther.sanchez@unison.mx (E.S.-V.); hernan.celaya@unison.mx (H.C.-M.); andres.ochoa@unison.mx (A.O.-M.); 2Centro de Investigación en Alimentación y Desarrollo, A.C. (CIAD, A.C.), Carretera a la Victoria Km. 0.6, Hermosillo 83304, Mexico; ecorral120@estudiantes.ciad.mx (E.A.C.-M.); martin@ciad.mx (M.V.-M.); libertad@ciad.mx (L.Z.-G.)

**Keywords:** growth promoters, pork production, chromium methionine, heat stress

## Abstract

**Simple Summary:**

This study aimed to evaluate the effects of the dietary supplementation of chromium methionine (CrMet) and ractopamine (RAC) on the productive behavior, blood metabolites, organ weight, carcass characteristics, and meat quality of pigs under heat stress. This study was conducted in the northwest of Mexico, where pig production is considerably affected during summer owing to heat stress. Supplementation with CrMet and RAC reduces the negative effects of heat stress and improves the productive behavior, blood components, and meat quality of livestock. However, the use of RAC has been restricted in several countries due to its potential negative health effects. Our results indicated that RAC supplementation improved the productive behavior and carcass characteristics of pigs compared with the control group. Likewise, dietary supplementation to pigs with 0.8 ppm of chromium from CrMet during the last 34 days of finishing produced a moderate increase, exhibiting similar results to RAC in weight gain, carcass quality, blood components, and meat quality. Thus, CrMet supplementation could be an alternative to use as a growth promoter.

**Abstract:**

This study aimed to determine the effects of the dietary supplementation of chromium methionine (CrMet) and ractopamine (RAC) on pigs in the growing–finishing stage under heat stress. The parameters evaluated included productive behavior, blood components, carcass characteristics, organ weight, and meat quality. This study was conducted during the summer season in Sonora, Mexico. The treatments included: (1) control diet (CON), a base diet (BD) formulated to satisfy the nutritional requirements of pigs; (2) RAC, BD plus 10 ppm RAC supplemented during the last 34 days of the study; (3) CrMet-S, BD supplemented with 0.8 ppm of Cr from CrMet during the last 34 days; and (4) CrMet-L, BD supplemented with 0.8 ppm of Cr from CrMet for an 81 d period. RAC supplementation improved the productive behavior and main carcass characteristics of the pigs compared with CON. However, RAC and CrMet supplementation during the last 34 days showed similar results in terms of weight gain, carcass quality, blood components, organ weight, and meat quality. The addition of CrMet-S had a moderate (although not significant) increase in productive performance and carcass weight. These findings are encouraging, as they suggest that CrMet may be a potential alternative for growth promotion. However, more research is needed.

## 1. Introduction

Global warming has led to an increase in environmental temperatures, which have been estimated to continue to rise worldwide [1,2]. High temperatures induce heat stress in pigs, due to which the pork industry reportedly loses millions of dollars annually [3]. This loss of income can be primarily explained by a reduced and inconsistent growth performance of the pigs, a decrease in the price of carcasses due to their weights and fat deposition, and an increase in animal mortality, especially among pigs destined for the market [4,5].

Reportedly, heat stress affects the behavior, physiology, and metabolism of pigs [6], leading to higher fat deposition in the tissues and lower muscle mass than what is expected according to the feed intake of these animals [4,6,7]. Consequently, this abnormally increases circulating insulin concentrations, which is one of the biochemical events that lead to physiologic and anatomic changes [7]. Therefore, new dietary approaches that help improve the nutritional metabolism of pigs reared under heat stress conditions must be established [8].

Ractopamine hydrochloride (RAC) is an important β2-adrenergic agonist that helps improve protein synthesis and promotes fat breakdown (lipolysis) [9]. Furthermore, RAC supplementation reportedly helps improve productive behavior and carcass characteristics in pigs reared under heat stress conditions [10,11]. However, the use of this agonist is currently questionable owing to the global market trend, and it is banned in different countries due to its potential detrimental effects on human health [12,13,14]. Therefore, the pork industry is seeking a growth-promoting compound that can replace RAC [15].

Chromium (Cr) supplementation can improve certain productive parameters in farm animals under heat stress conditions [16]. Therefore, dietary Cr supplementation could be an alternative to RAC for pigs as Cr promotes the action of insulin on glucose, proteins, and fat metabolism in animal tissues [17], thereby increasing muscle mass and decreasing fat percentage, which further improves meat quality [18].

Inorganic and organic Cr sources are used in animal production. The most commonly used inorganic Cr sources are Cr sulfate and Cr chloride [19]. However, recently, the use of organic Cr sources, such as Cr picolinate, Cr yeast, Cr propionate (CrProp), and Cr methionine (CrMet), has reinitiated [20], as they are considered more bioavailable than the inorganic Cr sources. CrMet exhibits the greatest potential to replace RAC supplementation from the aforementioned organic Cr sources, owing to its higher bioavailability than the other Cr chelates. This is because CrMet is chelated with an amino acid and, hence, it can easily cross the intestinal cell membrane and metabolize without prior digestion [18,21].

Furthermore, previous studies regarding farm animals have reported that dietary CrMet supplementation exerts beneficial effects on organ weight, feed conversion, carcass characteristics, meat quality, and blood metabolites [22,23]. However, despite the beneficial effects of CrMet on animal production, the effects of its dietary supplementation on pigs in the growth–finishing stages reared under heat stress conditions, and whether it can replace RAC, remain unknown.

Therefore, we hypothesized that CrMet supplementation for a long or short period (81 or 34 d, respectively) during the summer season improves the productive behavior, blood metabolites, carcass characteristics, and meat quality of pigs, similar to the effects of RAC. Therefore, this study aimed to evaluate the effects of dietary CrMet supplementation from the growth stage (81 d) or only at the end phase (34 d) on the productive behavior, blood metabolites, organ weight, carcass characteristics, and meat quality of pigs under heat stress conditions.

## 2. Materials and Methods

All animal procedures were performed according to the Mexican norms of animal care [24,25]. The study was approved by the University of Sonora Ethics Committee (Approval No.: USO313006286).

### 2.1. Housing and Animal Conditions

The study was conducted in the swine experimental unit of the Department of Agriculture and Livestock, University of Sonora (northwestern Mexico), which has an open-sided, barn-like commercial farm. Temperature and relative humidity were recorded daily during the feeding test, and the temperature humidity index (THI) was calculated based on these readings to understand the conditions and level of the heat stress to which the animals were exposed. The THI was obtained using the following formula: THI = [0.81 × T] + [(HR/100) × (T – 14.4)] + 46.4, where T is the temperature in °C and RH is the relative humidity in % [26].

Fifty-six twelve-week-old pigs (twenty-eight males and twenty-eight females) of terminal Yorkshire × Duroc crosses, with an average live weight of 43 kg, were included in this study. The animals were identified with tags and housed in individual pens equipped with stainless steel feeders and a nipple drinker. Food and water were provided ad libitum, and the animals were fed in three phases (Table 1): phase I (40–70 kg live weight), phase II (70–95 kg live weight), and phase III (95–130 kg live weight). Four experimental groups (with each one containing fourteen animals, *n* = 14) were formed with uniformity in the live weight of animals and similar proportions of gender per group.

### 2.2. Experimental Diets and Periods

Each pigs group was randomly assigned to one of the four following experimental diets: (1) control diet (CON), which included a base diet (BD) formulated to satisfy the nutritional requirements of the pigs with high lean potential and productivity [27]; (2) RAC, BD supplemented with 10 ppm RAC during the last 34 days of the finishing period (days 47–81 of feeding); (3) CrMet-S, BD supplemented with 0.8 ppm of Cr from CrMet during the last 34 days of feeding (days47–81); and (4) CrMet-L, BD supplemented with 0.8 ppm of Cr from CrMet for the entire evaluation period (days 0–81).

### 2.3. Source of CrMet and RAC

The brand name of the CrMet (Cr 2000 ppm) used in this study was BIOWAYS Cr^®^, supplied by Grupo Biotecap, S.A, de C.V. (Tepatitlán, Jalisco, México); the RAC utilized in this study was the commercial product PAFMINE (Laboratorios Paffa S.A de C.V., Popo Park, México).

### 2.4. Productive Performance

Throughout the study period, the total weight of the food supplied and rejected by each animal was recorded daily. The evaluation periods included the complete period of 81 days (days 1–81) and the last 34 days (days 47–81). The feed intake (FI) was calculated based on the difference between the daily feed served and rejected. Individual live weight was recorded at the beginning and end of each feeding period, and the average daily gain (ADG) was estimated using these data. Feed conversion (FC) was calculated based on the relationship between the FI and the ADG.

### 2.5. Blood Components

Blood samples were collected from 10 pigs per treatment (~14 mL of blood was collected via jugular venipuncture) on days 47 and 81 using two Vacutainer tubes (Becton, Dickinson and Company, Franklin Lakes, NJ, USA). One tube contained ethylenediaminetetraacetic acid (EDTA) and the other tube did not contain any additives. The tubes containing EDTA and blood were used for a complete blood count where red blood cells, hemoglobin, hematocrit, leukocytes, mean corpuscular volume (MCV), mean corpuscular hemoglobin (MCH), mean corpuscular hemoglobin concentration (MCHC), lymphocytes, and platelets were counted. The tubes without additive were centrifuged at 10,000 rpm for 10 min. The serum was separated and stored at −20 °C until the blood glucose, total protein, albumin, creatine kinase (CK), and cortisol parameters were analyzed. These parameters (except for cortisol) were quantified using the corresponding laboratory kits following the manufacturer’s instructions (RANDOX Manual). The cortisol levels were determined using the enzyme-linked immunosorbent assay (Sigma-Aldrich, St. Louis, MO, USA, EE. UU).

### 2.6. Slaughter and Carcass Characteristics

The pigs were slaughtered once the feeding trial was completed in the slaughterhouse of the Department of Agriculture and Livestock, University of Sonora. The pigs were electrically stunned before puncturing and exsanguinating the animals following the procedures established in the current official Mexican standards [25]. Hot carcass weights were recorded, and then the carcasses were cooled for 24 h at 4 °C to obtain the cold carcass weights; the carcass length was also measured, and hot and cold carcass yield percentages were estimated. Fat thickness (mm), *longissimus thoracis* (LT) muscle area (cm^2^), and marbling were determined on the left side of the carcass at the level of the 10th and 12th intercostal space (10-point subjective scale [28]).

### 2.7. Percentage Relationship between Organ and Live Weights

The liver, heart, lungs, stomach, spleen, and kidneys were removed from each pig, and the organ weights were recorded to calculate the percentage ratio between the organ weight and live weight of the pigs.

### 2.8. Dissection of the Longissimus Thoracis Muscle

Once the carcass characteristics were evaluated at 24 h *postmortem*, the portion of the LT muscle was extracted from the left side of the carcass between the 4th and 12th intercostal spaces. Then, the samples were vacuum packed and transported under refrigeration to the Meat Science and Technology Laboratory of the Centro de Investigación en Alimentación y Desarrollo (CIAD) in Hermosillo, Sonora, México, for subsequent meat quality analysis.

### 2.9. Meat Quality

#### 2.9.1. Chemical Composition and Physicochemical Characteristics

The moisture, intramuscular fat, and protein content of the LT muscle were determined following the AOAC methods [29] for moisture (method 950.46), fat (method 920.39), and protein (method 955.04). The results obtained were expressed as a percentage of the fresh weight.

The pH of the cold meat samples (4–6 °C) was determined using a HANNA portable digital potentiometer (Hanna Instruments, Woonsocket, RI, USA) with a penetrating electrode fitted with a HANNA HI 99163 thermometer. The measurements were made in triplicates.

The water holding capacity (WHC) was measured in the meat according to a previously established methodology [30]. The sample was placed on a micro nylon cloth and introduced into a 50 mL propylene tube. Then, the sample was centrifuged at 2800× *g* for 5 min at 4 °C. The WHC percentage was calculated based on the difference in the weight of the sample before and after centrifugation.

The texture of the meat was measured using a texturometer (Texture Analyzer TAXT-Plus, Texture Technologies Corp., Scarsdale, NY, USA). To measure the shear force of the meat, 3 cm-thick steaks were sectioned and cooked on an electric grill (Cook Master Ester, Model 3222-3, Mississauga, ON, Canada) until an internal temperature of 71 °C was reached. The samples were then cooled to room temperature (25 °C) and then refrigerated at 4 °C for 24 h. Subsequently, the cooked samples were sectioned into pieces of 1 cm by 3 cm along the direction of the muscle fibers (10 pieces/sample) and were measured perpendicularly to the muscle fibers using the Warner–Bratzler cutter accessory mounted on the texturometer. The Warner-Bratzler shear force (WBSF) values were expressed in kilograms.

The cooking loss (%) was calculated based on the difference in the weight of the sample before and after cooking to evaluate its texture, according to the American Meat Science Association (AMSA) technique [31].

#### 2.9.2. Fatty Acid Profile

The lipid fraction of each sample was recovered according to the procedure described by Bligh and Dyer (1959), with some modifications [32]. Briefly, 2 g of meat was mixed with 10 mL of methanol and homogenized (Ultra Turrax T25, IKA, Staufen, Germany) for 1 min at 11,000 rpm. Then, 20 mL of chloroform with a purity of 98.5% was added to the mixture and homogenized at 8000 rpm for 2 min. Subsequently, the samples were filtered through Whatman #1 paper, and 3 mL of KCl (0.88%) was added to the filtrate immediately following this step, and the mixture was vigorously shaken to obtain the upper phase. Finally, the mixture was washed with 2 mL of distilled water and 2 mL of methanol. The lower phase (chloroform and lipids) was collected, and the upper phase was removed using a Pasteur pipette for later use. Finally, the air in the vials was removed using nitrogen flow (Industrial Grade, PRAXAIR, Veracruz, Mexico), which were stored under refrigerated conditions from 4 °C to −18 °C in a concave tube for their esterification.

Fatty acid methyl esters (FAMEs) were prepared according to the technique described by Park and Goins [33], with some modifications. The lipid extracts were placed in a water bath at 40 °C, and the solvent was evaporated from the extracts via nitrogen flow. Approximately 150 mg solvent-free fat was obtained, to which 4 mL of 0.5 N NaOH in methanol was added. The mixture was lightly shaken until a homogeneous solution was obtained. The air was evacuated using nitrogen flow injection, and the tube was sealed with a Bakelite stopper and subsequently heated at 90 °C for 10 min in a water bath. Then, the samples were cooled to 25 °C, to which 5 mL of a 14% solution of boron trifluoride in methanol was added.

The samples were heated again in a water bath at 90 °C for 7 min. After cooling them to room temperature, 4 mL of heptane was added, and the samples were heated for 2 min under the abovementioned conditions. Subsequently, the samples were left to cool to room temperature, following which 1 mL of saturated sodium chloride (37 g NaCl/100 mL of distilled water) and anhydrous sodium sulfate were added. Finally, the upper crystalline phase was exclusively collected and placed in a vial for storage. At the time of sample injection, 100 µL of sample plus 100 µL of heptane was poured into a microinsert in a 2 mL vial for chromatography.

The FAMEs were analyzed using a gas chromatograph (Hewlett Packard Series 6890, Waldbronn, Germany) fitted with a flame ionization detector and silica capillary column (Agilent J&W DB-23, 0.25 mm internal diameter × 60 m length, with a film thickness of 0.25 μm, Agilent, Penang, Malaysia).

The oven temperature program included the following cycles: 120 °C at the start; an increase of 5 °C/min till reaching 175 °C, which was maintained for 1 min; and a second increase of 4 °C/min till reaching 230 °C, which was maintained for 10 min. The temperature of the injection port and detector were maintained at 250 °C and 260 °C, respectively. The samples were injected (1 µL) into a split port under a division ratio of 50:1. Helium was used as a transport gas with a speed of 14 cm/s, and the chromatograms were recorded and stored on the computer using Chemstation software LTS 01.11.

The fatty acids were identified according to the retention time and pattern of elution of the FAME Supelco Standards Mix C4-C24 (Sigma-Aldrich, St. Louis, MI, USA). The values were expressed as a percentage of the total quantity of FAMEs detected. The partial sums of saturated fatty acids (ΣSFA), monounsaturated (ΣMUFA), polyunsaturated (ΣPUFA), omega-3 (Σw3), and omega-6 (Σw6) were calculated. The nutritional ratios ΣMUFA/ΣSFA, ΣPUFA/ΣSFA, and Σw6/Σw3 were also calculated.

#### 2.9.3. Refrigeration Stability Study at 4 °C

Two chops of meat per experimental unit were packaged using the traditional shrink wrap and stored at 4 °C in a cooling chamber for 7 days in the presence of light to conduct the refrigeration stability study and simulate the commercial conditions of sale. The evaluation of the color, and lipid and protein oxidation of the meat was performed on days 0 and 7 of refrigeration. For the color evaluation, a CR-2600 Minolta colorimeter was used with D65 lighting, with 10 degrees as the observer and 8 mm as the opening diameter (Konica Minolta Sensing, Inc., Osaka, Japan). The parameters L* (lightness), a* (redness), and b* (yellowness) were employed for this evaluation. The Hue angle was calculated using the following formula: Tan^−1^ (b*/a*). Color determination was conducted in five different sites on the surface of the cold samples (4–6 °C) [34].

Lipid oxidation was evaluated using the thiobarbituric acid reactive substances (TBARS) technique, which determines lipid oxidation by measuring malonaldehyde (MDA) concentration. To achieve this, 5 g of sample was homogenized with 15 mL of trichloroacetic acid at 11,000 rpm for 1 min (Ultra Turrax IKA model T25, DEU, Staufen, Germany). Then, the homogenized sample was centrifuged at 592× *g* for 30 min at 5 °C (Thermo Scientific Legend XTR refrigerated centrifuge, Waltham, MA, USA). The supernatant was filtered through Whatman #42 paper following centrifugation. Then, 2 mL of the filtrate was collected, to which 2 mL of 20 mM thiobarbituric acid was added. The sample was subsequently homogenized for 30 s and heated in a water bath at 97 °C for 20 min. Finally, the tubes were cooled, and the absorbance of the samples was measured at 532 nm using a spectrophotometer (Cary 60 UV–Vis, Agilent Technologies, Bayan Lepas, Penang, Malaysia). From the absorbance, the MDA content was calculated and expressed in mg/kg of sample [35].

The procedure for measuring the percentage of metmyoglobin (MetMb) was as follows: 20 mL of cold phosphate buffer (pH = 6.8; 40 mM) was added to 5 g of fresh meat and homogenized at 11,300 rpm for 30 s (Ultra Turrax IKA model T25). Then, the sample was stored at 4 °C for 1 h, following which it was centrifuged at 2800× *g* for 30 min at 4 °C (Thermo Scientific Legend XTR refrigerated centrifuge). Then, the supernatant was filtered through Whatman #1 paper, and the absorbance of the filtrate was measured at different wavelengths (700, 572, and 525 nm) using a spectrophotometer (Cary 60 UV-vis, Agilent Technologies, Bayan Lepas, Penang, Malaysia). The MetMb content was calculated based on the absorbances and expressed as a percentage.

### 2.10. Statistical Analysis

All data were analyzed using SAS statistical software (see 9.1. SAS Inst. Inc., Cary, NC, USA) [36]. Each pig was considered as an experimental unit. The variables of productive performance, blood metabolites, carcass characteristics, and meat quality were analyzed using analysis of variance (ANOVA GLM) to a completely randomized block design, with the initial weight being considered as a blocking factor. The data of the refrigeration stability study at 4 °C were analyzed under a completely random design with a 4 × 2 factorial arrangement, where the treatments and storage time constituted the first and second factors, respectively. Means were compared using Tukey’s honest significant test when a significant effect of the treatments was observed A *p* < 0.05 was considered statistically significant for all analyses.

## 3. Results

### 3.1. Climatic Conditions

During the entire experiment, the average temperature inside the farm was 30.38 °C ± 1.8 °C (Table 2). The maximum and minimum temperatures recorded were 37.53 °C ± 2.5 °C and 24.80 °C ± 1.7 °C, respectively. The average humidity was 45.23% ± 4.8%. An average THI of 78.26 ± 3.6 was obtained with this information, with a maximum of 93.39 ± 4.8 and a minimum of 68.63 ± 10.7, respectively.

### 3.2. Productive Performance

The results of the productive performance per evaluation period are shown in Table 3. The final weight of the animals with the RAC treatment increased by 8.3% in comparison to the CON treatment (*p* < 0.05); however, there were no significant differences observed between the animals that were supplemented with CrMet-S and CrMet-L (*p* > 0.05).

When analyzing the entire period (81 days), it was determined that the pigs in the RAC group exhibited a greater ADG (12.62%) compared with those in the CON group (*p* < 0.05); however, RAC supplementation produced a similar ADG to that produced by CrMet (*p* > 0.05). There were no significant differences observed in the rest of the productive behavior variables (*p* > 0.05).

In the last period of evaluation (the last 34 days), the animals supplemented with CrMet-S exhibited a similar ADG to that of the animals supplemented with RAC (*p* > 0.05), and an increase of 19.23% and 16.98% was observed in the ADG of the RAC group compared with the CON and CrMet-L groups, respectively (*p* < 0.05). No significant differences were observed in the remaining parameters among the different treatment groups (*p* > 0.05).

### 3.3. Blood Components

The blood samples were collected on days 47 and 81. On day 47, none of the blood components were affected by the treatments (*p* > 0.05; Appendix A). However, significant differences were only found in the parameters of total proteins, albumin, and CK at the end of the feeding test (d 81) (*p* < 0.05; Appendix A). For the total proteins, the pigs supplemented with CrMet-S presented the same levels of total proteins as those who received the CON and RAC diets (*p* > 0.05); however, the total protein levels were reduced by 9.08% (*p* < 0.05) in the pigs supplemented with CrMet-L compared with the pigs supplemented with RAC.

Regarding the blood albumin levels, it was determined that the pigs who received CrMet-S and CrMet-L supplementation had the same level of blood albumin as the pigs who received RAC supplementation. However, the albumin levels in the blood were found to be significantly lower by 7.73% in the pigs that received the CON diet compared with the pigs that received RAC supplementation (*p* < 0.05). For CK, it was observed that the pigs in the CON, CrMet-S, and CrMet-L groups had significantly lower CK levels than the pigs in the RAC group (61.93%, 53.43%, and 51.86%, respectively, *p* < 0.01).

### 3.4. Carcass Characteristics

The carcass characteristics carcass are shown in Table 4. The carcass weight (hot and cold) was similar between the pigs supplemented with CrMet-S and RAC (*p* > 0.05); however, the hot and cold carcass weights in the CON and CrMet-L groups were found to be significantly lower by 8.3% and 6.7% compared with those in the RAC group, respectively (*p* < 0.05). As for the depth of the loin, the values were similar among the CON, CrMet-S, and RAC groups; however, the carcasses of the RAC group exhibited a greater depth of the loin than those of the CrMet-L group (*p* < 0.05). The loin area at the 12th rib space presented a tendency to be different (*p*< 0.10).

### 3.5. Percentage Relationship between Organ and Live Weights

There was no significant effect of the treatments observed on the relative weights of the organs (such as the liver, spleen, heart, lung, stomach, and kidneys) with respect to the live weight of the pigs (*p* > 0.05) (Appendix A).

### 3.6. Meat Quality

#### 3.6.1. Chemical Composition and Physicochemical Characteristics

Appendix A shows the results of the chemical and physicochemical characteristics of the meat. None of the evaluated parameters was affected by the treatments (*p* > 0.05).

#### 3.6.2. Fatty Acid Profile

Table 5 shows the FA profile of the intramuscular fat in the LT muscle of the supplemented pigs. No significant effect of the experimental diets on the content of any individual fatty acid (*p* > 0.05) was observed, except for on the C20:4ɷ6 FA content (*p* < 0.05). Specifically, the C20:4ɷ6 (arachidonic acid) content increased by 62.5% in the meat of the RAC group compared with that of the meat in the CON group. On the other hand, the C20:3ɷ6 FA content presented a tendency to be different (*p* < 0.10).

The sums of FAs and their nutritional relationships are shown in Table 6. A significant supplementation effect (*p* < 0.05) was observed on the sums of polyunsaturated FA (ΣPUFA) and omega 6 (Σn-6), where both variables were increased in the RAC group (33.8% for ΣPUFA and 35.9% for Σn-6, respectively) than in the CrMet-S group. The treatments also affected the PUFA/SFA ratio (*p* < 0.05); this nutritional ratio was 47.8% higher in the RAC treatment than in the CrMet-S treatment; furthermore, in both the CON and CrMet treatments, the ratio was like those of the RAC and CrMet-C treatments (*p* > 0.05). Finally, the omega 6/omega 3 (*n*-6/*n*-3) ratio increased by 15.9% in the RAC group in comparison to the CON group (*p* < 0.05). The sums ∑SFA, ∑MUFA, and ∑*n*-3, as well as the MUFA/SFA ratio, did not differ significantly between the dietary supplementations of CrMet and RAC (*p* > 0.05).

#### 3.6.3. Refrigeration Stability Study at 4 °C

Table 7 shows the results of the refrigerated stability study. Refrigeration time affected all the parameters (L*, a*, b*, Hue, and TBARS; *p* < 0.01). For the L* parameter, CrMet-S supplementation significantly increased the L* values compared with the rest of the treatments (*p* < 0.05); the L* values in the CrMet-S group was 9.52%, 8.40%, and 9.23% higher compared with those in the CON, RAC, and CrMet-L groups on day 1, respectively. Likewise, on day 7 of storage, in the CrMet-S group, the L* value was 9.82%, 5.16%, and 4.81% higher compared with those in the CON, RAC, and CrMet-L groups, respectively. These treatments had a tendency to be different in the values of b* and MetMb (*p* < 0.10).

## 4. Discussion

### 4.1. Climatic Conditions

High environmental temperatures cause heat stress in animals, which negatively affects their welfare and productivity, especially during the summer period. Consequently, heat stress constitutes one of the main challenges in pig farming [37], especially in the warmer parts of the world, such as Sonora in Mexico, where this study was conducted. During the entire experiment, the average temperature within the experimental farm was 30.38 °C ± 1.8 °C, and the average maximum and minimum temperatures recorded were 37.53 °C ± 2.5 °C and 24.80 °C ± 1.7 °C, respectively. Furthermore, the thermoneutral zone for growing and finishing pigs is between 18 °C and 25 °C [38], which is the environmental temperature range within which pigs do not have to use additional energy to maintain their body temperature and their productive functions develop optimally (free of stress) [39]. As heat stress occurs when the ambient temperature is higher than the thermoneutral zone of the animal, this indicates that in this study, the pigs were under heat stress conditions most of the time. Regarding the THI, the average THI was 78.25, while the maximum and minimum THIs were 93.39 and 68.63, respectively. According to the THI data obtained in this study and comparison with the scale determined by Hahn, the pigs were determined to be under alert and danger conditions for most of the time [26].

### 4.2. Productive Behavior

Heat stress in pigs lowers the ADG, FI, and FC [7,8], thereby decreasing in energy, as well as other nutrients for tissue synthesis [16,40], which, in part, explains the decrease in productive behavior. Consequently, the effects of CrMet supplementation were evaluated and compared with the effects of RAC supplementation during the summer period on the ADG, FI, and FC of pigs reared under heat stress conditions.

Notably, when assessing the complete study period (81 d) in terms of productive behavior, no significant differences were found between the groups of pigs that received a diet supplemented with CrMet-S, CrMet-L, and RAC, suggesting an approximation of the effects of CrMet to those of RAC on productive behavior. However, improvement in the ADG and final BW of the pigs could be attributed to an increase in protein synthesis in the tissue of their skeletal muscle induced by organic Cr (CrMet) [11,15,41].

During the last period (d 34 and 47–81), the results indicated that the pigs receiving a diet supplemented with CrMet-S and RAC exhibited a similar ADG. The ADG was increased in the RAC group than in the CON and CrMet-L groups. Furthermore, in terms of the pig final BW, similar final weights were obtained with supplementation in the CrMet-S, CrMet-L, and RAC groups.

These results are consistent with those reported by other studies that compared the effects of dietary organic Cr and RAC supplementation in finishing pigs [42,43,44]. Furthermore, Mayorga et al. reported that the addition of organic Cr (CrProp) in the diets of pigs reared under heat stress conditions exerted positive effects on the ADG and final BW [8].

Conversely, other studies reported that RAC supplementation improved the ADG and final BW compared with organic Cr supplementation [11,45]. The different results reported by these studies may be attributed to the various conditions under which the experiments were conducted, such as the type of diets, level of CrMet or RAC, type of organic Cr, addition time, and environment.

### 4.3. Blood Components

Blood biochemistry (hematological profile) is a rapid analysis method regularly used in clinical practice for various animals [46] and can be used as an indicator to determine heat stress [47], as some blood components change when animals are under heat stress conditions [47,48]. When studying the effects of the dietary supplementation on the pigs using their hematological profiles at the 47th day of the study, no change was observed in their blood metabolites, which may have been possibly due to the duration of CrMet supplementation, animal weights, and stress severity due to the heat the pigs were submitted to during the study. Likewise, Liu et al. reported that the effects of dietary Cr supplementation may depend on the magnitude of the heat stress that the pigs are submitted to [49].

At the end of the investigation (d 81) the levels of the total proteins and albumin in the blood were analyzed, revealing similar levels in the RAC and CrMet-S groups. This corroborates the capacity of CrMet to synthesize protein in the muscle tissue of pigs under heat stress conditions as the total proteins and albumin in the blood are involved in the protein metabolism of pigs [46,50,51]. Another important finding was that CK levels increased when the pigs were fed a diet supplemented with RAC, differentiating it from the rest of the treatments. CK is an enzyme mainly found in the heart and skeletal muscle, and its presence in the blood indirectly indicates muscle damage [52,53]. The highest values of albumin and CK coincided with the treatment that obtained the greatest weight gain, which shows that these parameters are closely linked to the growth of the animals. Therefore, it is undesirable for animals to have high levels of CK, and as RAC supplementation increases CK, this indicates that its supplementation to pigs destined for the market can be harmful not only for the pigs, but also for human health. Thus, future studies need to be conducted in this respect to confirm these results.

### 4.4. Carcass Characteristics

Carcass characteristics are important for the profitability of pork meat producers as the price of the carcass is adjusted according to these components, particularly for back fat thickness and loin depth [54]. However, carbohydrate, lipid, and protein metabolism undergo changes when pigs are under heat stress conditions [8], which are consequently reflecting in carcasses having more fat and less skeletal muscle due to a reduction in the movement of lipids that are reportedly mediated by hyperinsulinemia [7,49]. Thus, the characteristics of the pig carcasses under heat stress conditions were assessed. There were no significant differences in the weight of the carcass (cold and hot) and loin depth (an approximate indicator of muscle accumulation) between the pigs supplemented with CrMet-S and RAC. RAC supplementation increased the carcass weights compared with the CON diet and CrMet-L supplementation. No variation was observed in the levels of fat among the treatment groups, suggesting that the back fat of pigs can be reduced when reared under heat stress conditions and supplemented with CrMet-S and RAC. This is because it is normally expected that the higher the weight of the pigs, the higher the proportion of backfat. Our results are consistent with those reported by Hung et al., who used a diet supplemented with nanotripicolinate for pigs under heat stress conditions [55]. The findings are also consistent with those of Marcolla et al., who reported no differences in the productive behavior of pigs, carcass characteristics, and back fat thickness; although, contrary to the results of the present study, they reported differences in the weight of the carcasses when comparing RAC and organic Cr supplementation [11].

Conversely, Farias et al., who conducted a similar study to ours, did not report differences in the loin area and back fat when comparing RAC and organic Cr supplementation [42], which the authors attributed to an improvement in the efficiency of the pigs to metabolize fats due to Cr supplementation. Moreover, Liu et al. reported that Cr supplementation could help improve metabolism, specifically of lipids, while reducing the body fat of pigs under heat stress conditions [49]. Therefore, the dietary supplementation of both RAC and CrMet help improve the metabolism of pigs reared under heat stress conditions. Notably, in this study, like the diets developed by Farías et al., Marcal et al., and Rincón-Folres et al., the concentration of amino acids did not increase, which normally occurs in RAC supplementation; however, the effects of RAC supplementation were considerably better than those of the CON diet [42,43,44]. The characteristics of the carcasses were closely related to the productive behavior, since they presented the same performance between the treatments, especially for the CrMet-S and RAC groups.

### 4.5. Percentage Relationship between Organ and Live Weights

Pigs reared under heat stress conditions exhibit reduced organ weights, which is an adaptation to reduce heat production from the metabolic processes of the visceral organs [56]. In addition, the use of growth-promoting molecules or certain diseases can cause organ growth, consequently affecting the final weight and productive behaviors of pigs. Thus, the percentage relationship between the organ weights and live weight of the pigs under heat stress conditions was determined. The results revealed no significant increase or abnormal reduction in the percentage relationship between organ and live weights due to RAC and CrMet supplementation. These results differ from those reported by Hamidi et al. and Van Hoeck, who found significant differences in the heart and liver weights following supplementing chicks with organic Cr for fattening [57,58]. Thus, these results confirm that CrMet or RAC supplementation under heat stress conditions, at selected doses, does not change the percentage relationship between the organ and live weights of pigs.

### 4.6. Meat Quality

#### 4.6.1. Physicochemical Analysis

Heat stress in pigs can change certain meat quality parameters, including pH, water holding capacity, and color, generating a lower acceptance by consumers [59,60]. Herein, no significant differences among the parameters of physicochemical quality were observed in the different treatment groups. These results are consistent with those of other studies that did not report differences when using supplements, such as Cr [19] or RAC [11].

#### 4.6.2. Chemical Composition

There were no significant differences uncovered from the proximal analysis (percentage of moisture, grease, protein, and ash), and these results are consistent with those reported by Marcolla et al., wherein organic Cr supplementation to pigs at the end of the study was compared with RAC supplementation [11]. However, notably, our results were within the normal parameters, as previously observed by different studies which involved pigs raised under heat stress conditions [61,62].

Regarding the FA profile, currently, consumers are more conscious of the benefits of consuming good quality fat in their daily diet, with a trend being detected in the preference for products with higher unsaturated FA contents, as this type of FA is considered beneficial for health. Thus, it is important to identify the concentration of different FAs present in foods. The lipid profile revealed that the main SFAs were palmitic acid (C16:0) and stearic acid (C18:0). Oleic acid (C18:1ɷ9c) was the most abundant MUFA, while linoleic acid was the most frequently detected polyunsaturated fatty acid.

RAC increased the FA C20:4ɷ6 content (*p* < 0.05), consistent with the findings of Trujillo et al., who observed changes in the concentration of PUFA in the meat of pigs supplemented with 10 ppm RAC and 0.4 mg organic Cr/kg of food [44]. Furthermore, Jin et al. and Tian et al. supplemented 0.2 mg CrMet/kg of food in the diet of pigs and reported that the concentration of arachidonic acid in the meat of pigs supplemented with CrMet did not change following the treatment [63,64]. Arachidonic acid is considered an essential FA and is extremely important in breastfeeding women. However, its supplementation is not necessary in healthy adults if the usual diet contains >2.5% linoleic acid [65,66]. Thus, higher FA contents in the meat (RAC) does not necessarily imply a higher quality of the product; however, the meat from the CON group and both Cr treatment groups presented values within the normal range in the lipid profile. Their numerical values were also considered to be adequate to cover the nutritional requirements of the diet of an adult person.

#### 4.6.3. Refrigeration Stability Study at 4 °C

As expected over several days, during storage, refrigeration of the meat affected the color a*, b*, hue angle, lipid oxidation (TBARS), protein oxidation (MetMb), and cooking loss, regardless of the diet previously assigned to the pigs. An effect was also observed on the L* parameter when using CrMet-S on the different days of storage. The parameter L* (known as luminosity, where a higher value indicates a more brilliant color) suggests that the meat of the pigs supplemented with CrMet-S on days 1 and 2 was more luminous and brilliant. The effect on the L* parameter of this study does not match the one reported by Marcolla et al., who supplemented organic Cr to pigs and found that the L* values in the meat were equal to that of those receiving RAC-supplemented and CON diets [11]. The results observed in the L* value due to storage time are consistent with those reported by Lindemann et al. [67].

Furthermore, reportedly, the b* value indicates the yellowness of meat and is used to determine the chemical status of myoglobin. Herein, the color parameter b* did not differ among the treatment groups, consistent with results reported by Almeida et al. [45]. González et al. and Lindemann et al. observed a treatment effect of higher supplemental doses of CrMet (0.5 and 0.8 mg/kg CrMet, respectively) and a decrease in the b* value of meat [67,68]. Moreover, herein, the meat hue angle of the pigs supplemented with Cr did not exhibit treatment effects; however, they did show storage time effects. Generally, the meat hue angle was inferior on day 1 to that observed on day 7 (41.76 vs. 64.93, respectively).

There were no treatment effects on the MetMb oxidation of the meat of the supplemented pigs; however, there were storage time effects, as lower values of MetMb percentages were obtained on day 1 than on day 7. According to the findings of Stanišić et al. (2016), who analyzed fresh LT samples from commercial and field pigs during different periods (days 1–6) of refrigeration at 4 °C ± 1 °C, a linear effect was observed between the storage days and increase in MetMb percentage [69].

Regarding lipid oxidation, the TBARS parameter of the meat of supplemented pigs showed storage time effects, whereupon at the end of storage (day 7), the meat exhibited superior lipid oxidation to that at the beginning of storage when the values were lower (1.067 at day 7 vs. 0.207 at day 1, respectively).

Finally, refrigeration stability study revealed that both CrMet and RAC supplementation did not considerably change the studied parameters, and although the pigs were subjected to heat stress conditions, the stability of the meat color was similar to that reported by other studies. Additionally, although there were no significant treatment effects on most of the evaluated parameters, the current information serves as a background to continue with the investigations on the possible benefits of CrMet supplementation on pork meat quality.

## 5. Conclusions

The results of this study showed that RAC had high productivity, and it was observed that the addition of CrMet in a short period (34 days) had a moderate (although not significant) increase in productive performance and carcass weight. These findings are encouraging, as they suggest that CrMet may be a potential alternative for growth promotion in finishing pigs. However, more research is needed to confirm these findings and determine the optimal dose and duration of supplementation. It is recommended that future studies should be conducted with a larger sample size and for a different time period.

## Figures and Tables

**Table 1 animals-13-02671-t001:** Composition of experimental diets for pigs in the growing–finishing stage.

Ingredients, %	Productive Phase (Live Weight)		
	Phase I (40–70 kg)	Phase II (70–95 kg)	Phase III (95–130 kg)
Corn grain	72.39	78.10	79.56
Soybean meal	22.82	17.50	16.09
Vegetable oil	1.79	1.60	1.55
Premix ^1^	3.00	2.80	2.80
Nutritional analysis			
Metabolizable energy, Mcal/kg	3.37	3.36	3.34
Crude protein, %	16.86	14.79	14.24
Digestible lysine, %	0.96	0.82	0.79
Digestible methionine, %	0.28	0.24	0.22
Calcium, %	0.69	0.68	0.66
Available phosphorus, %	0.30	0.23	0.23

^1^ Premix: premix of amino acids, vitamins, and minerals. Each kilogram of feed provided 9.5 g dicalcium phosphate, 8.3 g limestone, 3.55 g sodium chloride, 2.3 g l-lysine, 0.5 g dl-methionine, 0.35 g l-threonine, 0.15 g l-tryptophan, 8000 IU vitamin A, 800 IU vitamin D3, 40 IU vitamin E, 3.5 mg vitamin K, 7 mg riboflavin, 20 mg pantothenic acid, 30 mg niacin, 30 µg vitamin B12, 550 mg choline, 64 mg Zn, 64 mg Fe, 4 mg Cu, 4 mg Mn, 0.4 mg Y, and 13 mg Se.

**Table 2 animals-13-02671-t002:** Maximum temperature, minimum temperature, average temperature, relative humidity, and the temperature humidity index throughout the experimental period.

Value	Maximum	Minimum	Average
Temperature, °C	37.53 ± 2.5	24.80 ± 1.7	30.38 ± 1.8
Relative humidity, %	71.71 ± 13.7	33.67 ± 10.7	45.23 ± 4.8
Temperature humidity index	93.39 ± 4.8	68.63 ± 2.8	78.26 ± 3.6

The data are presented as means ± standard deviation.

**Table 3 animals-13-02671-t003:** Growth performance of pigs supplemented with chromium methionine in two periods.

	Treatment ^1^					
Parameter	CON ^1^	RAC ^2^	Met-Cr-S ^3^	Met-Cr-L ^4^	SEM ^5^	*p*-Value ^6^
Body weight						
IBW ^7^, kg	43.43	43.45	43.30	43.40	0.260	0.996
FBW ^8^, kg	127.60 ^b^	138.18 ^a^	132.49 ^ab^	130.21 ^ab^	1.170	0.020
Total period, 81 days (day 1 to day 81)						
ADG ^9^, kg/d	1.03 ^b^	1.16 ^a^	1.10 ^ab^	1.07 ^ab^	0.014	0.016
FI ^10^, kg/d	2.51	2.75	2.70	2.60	0.038	0.143
FC ^11^, kg/kg	2.42	2.36	2.46	2.43	0.029	0.666
Last period, 34 days (day 47 to day 81)						
ADG ^9^, kg/d	1.04 ^b^	1.24 ^a^	1.12 ^ab^	1.06 ^b^	0.021	0.011
FI ^10^, kg/d	2.89	3.19	3.12	3.00	0.050	0.170
FC ^11^, kg/kg	2.79	2.58	2.78	2.84	0.045	0.214

^1^ CON: control animals receiving only the basal diet without supplementation; ^2^ RAC: animals receiving the basal diet supplemented with 10 ppm ractopamine during the last phase (34 d); ^3^ Met-Cr-S: animals receiving the basal diet supplemented with 0.8 ppm of Cr from chromium methionine only during the last phase for a short period (34 d); ^4^ Met-Cr-L: animals receiving the basal diet supplemented with 0.8 ppm of Cr from chromium methionine used in all phases for a long period (81 d); ^5^ SEM: standard error of the mean; ^6^ probability values associated with the effects of chromium methionine or ractopamine supplementation in the diets of pigs; ^7^ IBW: initial body weight; ^8^ FBW: final body weight; ^9^ ADG: average daily gain; ^10^ FI: feed intake; and ^11^ FC: feed conversion. ^a,b^ Within rows, means with different superscript letters were significantly different (*p* < 0.05).

**Table 4 animals-13-02671-t004:** Carcass characteristics of the pigs under heat stress by treatment.

Parameter	Treatment					
	CON ^1^	RAC ^2^	Met-Cr-S ^3^	Met-Cr-L ^4^	SEM ^5^	*p*-Value ^6^
Hot carcass weight, kg	108.05 ^b^	117.80 ^a^	112.29 ^ab^	109.92 ^b^	1.136	0.011
Dressing, % (warm)	84.68	85.25	84.75	84.42	0.162	0.340
Cold carcass weight, kg	106.49 ^b^	116.24 ^a^	110.80 ^ab^	108.41 ^b^	1.134	0.012
Dressing, % (cold)	83.46	84.12	83.26	83.63	0.162	0.296
Length of carcass, cm	102.63	102.08	101.50	101.875	0.462	0.855
Backfat thickness 10th, cm	1.83	2.17	2.04	1.99	0.065	0.360
Backfat thickness 12th, cm	0.88	0.82	0.87	0.82	0.176	0.270
Loin depth 10th, cm	7.88	8.30	8.408	7.88	0.105	0.176
Loin depth 12th, cm	7.85 ^ab^	8.39 ^a^	8.18 ^ab^	7.64 ^b^	0.088	0.024
Loin area 10th, cm^2^	70.91	74.73	71.08	70.73	1.594	0.781
Loin area 12th, cm^2^	71.72	77.96	71.13	69.14	1.260	0.097
Marbling score	2.75	2.58	2.50	2.67	0.125	0.906

^1^ CON: control animals receiving only the basal diet without supplementation; ^2^ RAC: animals receiving the basal diet supplemented with 10 ppm ractopamine during the last phase (34 d); ^3^ Met-Cr-S: animals receiving the basal diet supplemented with 0.8 ppm of Cr from chromium methionine only during the last phase for a short period (34 d); ^4^ Met-Cr-L: animals receiving the basal diet supplemented with 0.8 ppm of Cr from chromium methionine used in all phases for a long period (81 d); ^5^ SEM: standard error of the mean; and ^6^ probability values associated with the effects of chromium methionine or ractopamine supplementation in the diets of pigs. ^a,b^ Within rows, means with different superscript letters were significantly different (*p* < 0.05).

**Table 5 animals-13-02671-t005:** Intramuscular fatty acid profile (%) of the *longissimus thoracis* muscle from pigs supplemented with chromium methionine or ractopamine under heat stress conditions.

	Treatment					
Fatty Acid, %	CON ^1^	RAC ^2^	Met-Cr-S ^3^	Met-Cr-L ^4^	SEM ^5^	*p*-Value ^6^
C10:0	0.07	0.06	0.09	0.07	0.010	0.460
C12:0	0.08	0.07	0.10	0.08	0.013	0.459
C14:0	1.39	1.28	1.44	1.40	0.051	0.249
C16:0	26.13	25.39	26.51	26.40	0.426	0.283
C16:1	3.10	2.79	2.94	3.02	0.110	0.264
C17:0	0.06	0.14	0.11	0.11	0.044	0.488
C17:1	0.05	0.08	0.07	0.09	0.022	0.689
C18:0	14.70	14.40	15.58	15.37	0.405	0.126
C18:1ɷ9c	42.03	40.67	41.53	41.15	0.711	0.614
C18:2ɷ6t	0.07	0.08	0.11	0.11	0.023	0.585
C18:2ɷ6c	8.55	10.04	7.99	8.51	0.651	0.143
C18:3ɷ3	0.43	0.45	0.41	0.41	0.034	0.791
C20:0	0.09	0.10	0.15	0.15	0.029	0.301
C20:1ɷ9	0.72	0.72	0.72	0.74	0.072	0.983
C20:2	0.36	0.42	0.33	0.37	0.033	0.403
C20:3ɷ6	0.25	0.38	0.23	0.29	0.042	0.080
C20:4ɷ6	1.52 ^a^	2.47 ^b^	1.27 ^a^	1.50 ^a^	0.175	0.001
C20:3ɷ3	0.01	0.06	0.05	0.04	0.024	0.548
C24:0	0.27	0.36	0.24	0.27	0.041	0.209

^1^ CON: control animals receiving only the basal diet without supplementation; ^2^ RAC: animals receiving the basal diet supplemented with 10 ppm ractopamine during the last phase (34 d); ^3^ Met-Cr-S: animals receiving the basal diet supplemented with 0.8 ppm of Cr from chromium methionine only during the last phase for a short period (34 d); ^4^ Met-Cr-L: animals receiving the basal diet supplemented with 0.8 ppm of Cr from chromium methionine used in all phases for a long period (81 d); ^5^ SEM: standard error of the mean; and ^6^ probability values associated with the effects of chromium methionine or ractopamine supplementation in the diets of pigs. ^a,b^ Within rows, means with different superscript letters were significantly different (*p* < 0.05).

**Table 6 animals-13-02671-t006:** Sums of fatty acids and their nutritional ratios of intramuscular fat from the *longissimus thoracis* muscle for each experimental treatment.

	Treatment					
Fatty Acid, %	CON ^1^	RAC ^2^	Met-Cr-S ^3^	Met-Cr-L ^4^	SEM ^5^	*p*-Value ^6^
∑SFA ^7^	42.82	41.83	44.25	43.88	0.772	0.122
∑MUFA ^8^	45.92	44.28	45.25	45.01	0.803	0.530
∑PUFA ^9^	11.25 ^ab^	14.04 ^b^	10.49 ^a^	11.05 ^ab^	0.871	0.026
MUFA/SFA	1.07	1.06	1.02	1.03	0.035	0.610
PUFA/SFA	0.26 ^ab^	0.34 ^b^	0.23 ^a^	0.25 ^ab^	0.020	0.019
∑*n*-3 ^10^	0.44	0.51	0.46	0.45	0.051	0.758
∑*n*-6 ^11^	10.36 ^ab^	13.01 ^b^	9.57 ^a^	10.11 ^ab^	0.842	0.025
*n*-6/*n*-3	23.60 ^a^	27.35 ^b^	21.02 ^a^	22.71 ^a^	1.041	0.029

^1^ CON: control animals receiving only the basal diet without supplementation; ^2^ RAC: animals receiving the basal diet supplemented with 10 ppm ractopamine during the last phase (34 d); ^3^ Met-Cr-S: animals receiving the basal diet supplemented with 0.8 ppm of Cr from chromium methionine only during the last phase for a short period (34 d); ^4^ Met-Cr-L: animals receiving the basal diet supplemented with 0.8 ppm of Cr from chromium methionine used in all phases for a long period (81 d); ^5^ SEM: standard error of the mean; ^6^ probability values associated with the effects of chromium methionine or ractopamine supplementation in the diets of pigs; ^7^ SAF: saturated fatty acids; ^8^ MUFA: monounsaturated fatty acids; ^9^ PUFA: polyunsaturated fatty acids; ^10^ *n*-3: omega 3 fatty acids; and ^11^ *n*-6: omega 6 fatty acids. ^a,b^ Within rows, means with different superscript letters were significantly different (*p* < 0.05).

**Table 7 animals-13-02671-t007:** Refrigeration stability at 4 °C of the parameters of color, lipidic, and protein oxidation of meat from pigs supplemented with chromium methionine or ractopamine under heat stress conditions.

	Variable					
Treatment	L*	a*	b*	Hue	TBARS ^7^	MetMb ^8^
Day 1						
CON ^1^	52.28 ^b^	6.88	6.26	41.76	0.245	43.54
RAC ^2^	52.82 ^b^	7.20	6.39	41.47	0.221	45.05
Met-Cr-S ^3^	57.26 ^a^	7.49	7.06	43.66	0.207	40.73
Met-Cr-L ^4^	52.42 ^b^	6.55	6.11	42.70	0.250	38.65
Day 7						
CON ^1^	54.78 ^b^	3.30	6.46	62.42	0.871	64.01
RAC ^2^	57.21 ^b^	3.39	7.19	64.88	0.703	64.82
Met-Cr-S ^3^	60.16 ^a^	4.09	7.51	64.93	0.783	63.23
Met-Cr-L ^4^	57.40 ^b^	3.72	7.51	63.89	1.067	61.70
SEM ^5^	1.590	0.493	0.522	2.331	0.126	1.843
*p*-Value ^6^						
Treatment	0.013	0.490	0.084	0.826	0.447	0.064
Day	0.001	0.0001	0.015	0.0001	0.0001	0.0001
Interaction	0.843	0.788	0.653	0.938	0.620	0.778

^1^ CON: control animals receiving only the basal diet without supplementation; ^2^ RAC: animals receiving the basal diet supplemented with 10 ppm ractopamine during the last phase (34 d); ^3^ Met-Cr-S: animals receiving the basal diet supplemented with 0.8 ppm of Cr from chromium methionine only during the last phase for a short period (34 d); ^4^ Met-Cr-L: animals receiving the basal diet supplemented with 0.8 ppm of Cr from chromium methionine used in all phases for a long period (81 d); ^5^ SEM: standard error of the mean; ^6^ probability values associated with the effects of chromium methionine or ractopamine supplementation in diets of pigs; ^7^ TBARS: thiobarbituric acid reactive substances (mg of malonaldehyde/kg tissue); and ^8^ MetMb: metmyoglobin (in percentage). ^a,b^ Within rows, means with different superscript letters were significantly different (*p* < 0.05).

## Data Availability

The data sets analyzed in the present study are available from the corresponding authors upon request.

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
