# Peer review of "Chromium Methionine and Ractopamine Supplementation in Summer Diets for Grower–Finisher Pigs Reared under Heat Stress"

_animals, 2023, doi:10.3390/ani13162671_

Round 1

Reviewer 1 Report

In this study, the authors determined the effects of the dietary supplementation of chromium methionine and ractopamine (RAC) on the productive behavior, blood metabolites, organ weight, carcass characteristics, and meat quality of pigs under heat stress. Key findings include that RAC improved the growth performance and carcass characteristics of pigs under heat stress. However, there are still major and minor concerns before it can be considered for publication.

1. what was the rationale for the chromium methionine and ractopamine dose and the adding time selected?

2. Please provide details on the age of the pigs as well as on the group size calculations.

3. How can the authors confirm pigs were under heat stress from the present results? As the author mentioned in the Discussion, organ weights will reduce in pigs reared under heat stress, what are the organ weights of pigs feeding in a normal environment? Why was there no group of pigs raised at a suitable temperature as a negative control?

4. From the results, compared with the CON group, the growth performance and carcass weight of pigs were not significantly improved by adding CrMet to the diet, could the authors explain the results?

5. Although there were no significant differences in growth performance and carcass characteristics between the Met-Cr addition groups and the RAC group, it could not be concluded that Cr-Met could be an alternative for RAC because the addition of CrMet did not significantly improve growth performance, and carcass characteristics compared with the CON group. The authors need to consider the accuracy of the conclusions

6. The form of all Tables should be uniform. (Table 1, line thickness)

Author Response

Thank you very much for reviewing and commenting on our manuscript. I am sending the responses to your comments and I am also attaching a word file where the modifications are explained. I'll be waiting for anything

thank you very much again

  1. what was the rationale for the chromium methionine and ractopamine dose and the adding time selected?

R: The doses of ractopamine and chromium methionine used in the experiment are those recommended by the manufacturer of both additives. For chromium methionine, it is 8 micrograms/kg (0.8 ppm) and for ractopamine a dose of 10 ppm is used commercially. Regarding the supplementation period, both additives are administered through the diet during the last 30 to 40 days of the finishing phase, prior to slaughter. Chromium supplementation for a long period was a proposal from our work group, hypothesizing that with this prolonged supplementation better effects on the productive performance of the animals would be obtained.

  1. Please provide details on the age of the pigs as well as on the group size calculations.

R: The age of the animals, 12-week-old;

The use of individual pens instead of group housing allows individual records to be made and, therefore, more precise information on the productive performance of the animals. In productive performance tests with pigs of the same commercial cross and with uniformity in live weight, it is possible to have a reduction in random and systematic errors, which allows minimizing the number of experimental units per group. It was considered to have at least 40 degrees of freedom in the error (in this study, 52 degrees freedom in the error) and ensure to detect real significances when they existed. Studies reported (including from our work group) for feeding trials with finishing diets with pigs housed individually, have detected statistical significance in productive performance using between 8 and 15 animals per group. Line 112 - 119

  1. (a) How can the authors confirm pigs were under heat stress from the present results? (b) As the author mentioned in the Discussion, organ weights will reduce in pigs reared under heat stress, what are the organ weights of pigs feeding in a normal environment? (c) Why was there no group of pigs raised at a suitable temperature as a negative control?

R: (a) Thanks for the observation. The environmental temperature and relative humidity were recorded, and with these data, the THI was estimated. The THI values estimated during the experimental period indicated that the animals were exposed to moderate to severe heat stress.

R: (b) There are studies that have evaluated organs weight in good conditions but these studies are not comparable to our study because may have presented different characteristics (ambient temperatures, different breeds, finals weights, installations, health status, etc.). We thought that adding certain components could affect the relationship of organs weights and live weight, it is known that organs are minor in proportion in summer, and we wanted evaluate if the additives used in our study could modify this relations (organ weights and live weight). We did not find differences; we thought that if we had found an improvement, we could have a better organ weight/live weight relationship.

R: (c) It would have been interesting to have a negative control group that was not exposed to heat stress conditions. However, the experimental site facilities are openly exposed to the prevailing environment during animal rearing, as is the case with commercial farms in the region. Therefore, there were no facilities with controlled environmental conditions.

  1. From the results, compared with the CON group, the growth performance and carcass weight of pigs were not significantly improved by adding CrMet to the diet, could the authors explain the results?

R: We think that it is due to the variation of the animals, this may be due to a number of factors, such as tolerance to weather conditions, health status, genetics, etc. (not detectable at the randomization). This can increase the variation within treatments and cause there to be no statistics differences.

  1. Although there were no significant differences in growth performance and carcass characteristics between the Met-Cr addition groups and the RAC group, it could not be concluded that Cr-Met could be an alternative for RAC because the addition of CrMet did not significantly improve growth performance, and carcass characteristics compared with the CON group. The authors need to consider the accuracy of the conclusions

R: Thanks, attended.   Line 712 - 718

  1. The form of all Tables should be uniform. (Table 1, line thickness)

R: The table format has been standardized.

Reviewer 2 Report

1.      L102-122: The animal condition and experimental design should be more detailed.

2.      Table 1: Why metabolizable energy were identical across different phases? Besides, dietary (digestible) Met level should be indicated.

3.      Footnote of Table 1: It is unknown that the presented contents of vitamins are vitamins themselves or their compounds? Besides, the mineral profile is missing.

4.      L120: How to determine the additive amount (0.8 ppm) of Cr?  

5.      L133: “Feed conversion (F:G)”, the abbreviation do not correspond to the full name.

6.      Table 3: Growth performance of pigs during the starter period (1-46 d) is suggested to be presented. Besides, the format of unit “Kg” is not correct. The same as below.

7.      Tables 5 and 6: SEM should be calculated to three decimal places according to other Tables. Besides, “EEM” in Table 6 is incorrect.      

8.      In discussion, the associations among different parameters should be referred and the discussion about different parameters should be integrated.

Author Response

Thank you very much for reviewing and commenting on our manuscript. I am sending the responses to your comments and I am also attaching a word file where the modifications are explained. I'll be waiting for anything.

Thank you very much again

Comments and Suggestions for Authors

  1. L102-122: The animal condition and experimental design should be more detailed.

R: More information regarding the experimental design was added. Line 112 - 119

  1. Table 1: Why metabolizable energy were identical across different phases? Besides, dietary (digestible) Met level should be indicated.

R:Thanks, it was corrected.   

  1. Footnote of Table 1: It is unknown that the presented contents of vitamins are vitamins themselves or their compounds? Besides, the mineral profile is missing.

R:Thanks, attended.   Line 121 -125

  1. L120: How to determine the additive amount (0.8 ppm) of Cr?  

R: The doses of ractopamine and chromium methionine used in the experiment are those recommended by the manufacturer of both additives. For chromium methionine, it is 8 micrograms/kg (0.8 ppm) and for ractopamine a dose of 10 ppm is used commercially. Regarding the supplementation period, both additives are administered through the diet during the last 30 to 40 days of the finishing phase, prior to slaughter. Chromium supplementation for a long period was a proposal from our work group, hypothesizing that with this prolonged supplementation better effects on the productive performance of the animals would be obtained.

  1. L133: “Feed conversion (F:G)”, the abbreviation do not correspond to the full name.

R: Thanks, attended.   Line 156, 500, 504

  1. Table 3: Growth performance of pigs during the starter period (1-46 d) is suggested to be presented. Besides, the format of unit “Kg” is not correct. The same as below.

R: For the first 46 days of the experiment, only the treatment with chromium long-period was added to the animals diet. The other groups received the basal diet. Therefore, we think it would not be meaningful compare the productive performance of the animals in this period.

´´the format of unit “Kg” is not correct´´ R: thanks, this was attended

  1. Tables 5 and 6: SEM should be calculated to three decimal places according to other Tables. Besides, “EEM” in Table 6 is incorrect.      

R: Thanks, attended.   

  1. In discussion, the associations among different parameters should be referred and the discussion about different parameters should be integrated.

R: Thanks, attended.   Lines 559 – 561; 597- 599;

Reviewer 3 Report

Report on the manuscript animals-2528490 entitled: Chromium methionine and ractopamine supplementation in summer diets for grower-finisher pigs reared under heat stress.

The authors carry out a study based on the hypothesis that CrMet supplementation for 34 and 81 days in summer in Mexico (Heat stress) might improve production parameters, animal behavior, blood metabolites, carcass characteristics, and meat quality of pigs, similar to the effect of RAC.

-          The authors mention that some studies regarding the use of CrMet can be found in the literature. Nevertheless, it is not stated or justified anywhere why the authors decided to study the duration of the supplementation and not the dose (CrMet concentration).

-          L. 44-54. Please, rewrite and be more concise and precise. The described facts are related to long-term HS. Short HS could lead to da different behaviour
In addition, the increase in circulating insulin is one of the biochemical events that lead to metabolic changes, etc.

-          L. 108. The authors mention “uniformity in sex”. Nevertheless, no information regarding the sex effect can be observed.

-          Table 1. How is it possible that the Met Energy remains constant?

-          L. 156-158 and table 4. Could the authors explain the procedure and units of Marbling evaluation?
It is mentioned USDA (2020). Scale or range? Units?

-          Table 4. Loin area 12th. I wonder whether such “tendency” should be considered or not.

-          Table 4. Please, review the Backfat thickness 12th values. SEM value is higher that Met-Cr-L value. If the values are correct, they indicate that Met-Cr-L analyses lack of accuracy…

-          Table 5. C20:3n-6. Again. I wonder whether such “tendency” should be considered or not.

-          Table 7. Same as above.

-          L. 164. “quality” or “carcass classification parameters”?

-          L. 166. Why were the samples “vacuum packed”? Was the lost water linked to the vacuum packing considered?

-          L. 187. “room temperature (25-30)” lacks of accuracy!

-          L. 187. Why “refrigerated for 24 h”?

-          L. 215. “room temperature” meaning?

-          L. 296. “behavior” is not appropriate. “Production parameters?”

-          L. 251. Chroma formula/equation is not correct.

-          L. 698-609. Please, rewrite being more accurate.

Extensive editing of English language required.

For example:, use:

Relate to

Compare to

Help to

Associate with

Author Response

Thank you very much for reviewing and commenting on our manuscript. I am sending the responses to your comments and I am also attaching a word file where the modifications are explained. I'll be waiting for anything

Thank you very much again

Comments and Suggestions for Authors

Report on the manuscript animals-2528490 entitled: Chromium methionine and ractopamine supplementation in summer diets for grower-finisher pigs reared under heat stress.

The authors carry out a study based on the hypothesis that CrMet supplementation for 34 and 81 days in summer in Mexico (Heat stress) might improve production parameters, animal behavior, blood metabolites, carcass characteristics, and meat quality of pigs, similar to the effect of RAC.

-          The authors mention that some studies regarding the use of CrMet can be found in the literature. Nevertheless, it is not stated or justified anywhere why the authors decided to study the duration of the supplementation and not the dose (CrMet concentration).

R: The doses of ractopamine and chromium methionine used in the experiment are those recommended by the manufacturer of both additives. For chromium methionine, it is 8 micrograms/kg (0.8 ppm) and for ractopamine a dose of 10 ppm is used commercially. Regarding the supplementation period, both additives are administered through the diet during the last 30 to 40 days of the finishing phase, prior to slaughter. Chromium supplementation for a long period was a proposal from our work group, hypothesizing that with this prolonged supplementation better effects on the productive performance of the animals would be obtained.

-     L. 44-54. Please, rewrite and be more concise and precise. The described facts are related to long-term HS. Short HS could lead to da different behaviour
In addition, the increase in circulating insulin is one of the biochemical events that lead to metabolic changes, etc.

R: Thanks, attended.  Line 53 - 60 

-          L. 108. The authors mention “uniformity in sex”. Nevertheless, no information regarding the sex effect can be observed.

R: Thanks for this observation. The sentence was changed to: four experimental groups (each one with 14 animals, n=14) were formed with uniformity in live weight of animals and similar proportions of gender per group. Lines 112-119

-    Table 1. How is it possible that the Met Energy remains constant?

R: Thanks, attended.   

-          L. 156-158 and table 4. Could the authors explain the procedure and units of Marbling evaluation?
It is mentioned USDA (2020). Scale or range? Units?

R: Thanks, attended.   Lines 181 - 182

NPPC, 1999

National Pork Producers Council (NPPC). 1999. Official color and marbling standards. NPPC, Des Moines, IA.

-          Table 4. Loin area 12th. I wonder whether such “tendency” should be considered or not.

R: Thanks, it was added Line 389

-          Table 4. Please, review the Backfat thickness 12th values. SEM value is higher that Met-Cr-L value. If the values are correct, they indicate that Met-Cr-L analyses lack of accuracy…

R: Thanks for this observation. The Backfat thickness value to Met-Cr-L is 0.82

-          Table 5. C20:3n-6. Again. I wonder whether such “tendency” should be considered or not.

      R: It was added Line 419

-          Table 7. Same as above.

      R: it was added Line 465

-          L. 164. “quality” or “carcass classification parameters”?

R: Quality word was changed to Carcass characteristics Line 188

-          L. 166. Why were the samples “vacuum packed”? Was the lost water linked to the vacuum packing considered?

R: Yes, the lost water generated by the vacuum packaging was considered to adjust the proximal composition, which was minimal because the transportation was approximately one hour.

-          L. 187. “room temperature (25-30)” lacks of accuracy!

R: The date was corrected, 25 °C. Line 215, 243.

-          L. 187. Why “refrigerated for 24 h”?

R: It is a time recommended in the protocol established by the American Meat Science Association (AMSA) for the evaluation of the WBSF texture.

-          L. 215. “room temperature” meaning?

R: It is the ambient temperature inside the laboratory (25 °C). This temperature was indicated. Line 243

-          L. 296. “behavior” is not appropriate. “Production parameters?”

R: Productive behavior was changed to “Productive performance”. Line 335, 336

-          L. 251. Chroma formula/equation is not correct.

R: An apology, the chroma values were not estimated or reported in the study. Only the Hue angle was estimated. Line 283 and Table 7.

-          L. 698-609. Please, rewrite being more accurate.

R: These lines are in the bibliography part, but we think you commented on the conclusion, we attended line 712 - 718

Comments on the Quality of English Language

Extensive editing of English language required.

For example:, use:

Relate to

Compare to

Help to

Associate with

R: We sent before, the manuscript to ENAGO for translation and writing services. This is the receipt: link  Translate and writing ENAGO.pdf

Round 2

Reviewer 3 Report

The authors have adequately carried out the review of their work.

--